# Lipid-DNA Nanoparticles as Drug-Delivery Vehicles for the Treatment of Retinal Diseases

**DOI:** 10.3390/pharmaceutics15020532

**Published:** 2023-02-04

**Authors:** Sven Schnichels, David Simmang, Marina Löscher, Andreas Herrmann, Jan Willem de Vries, Martin S. Spitzer, José Hurst

**Affiliations:** 1Centre for Ophthalmology, University Eye Hospital Tübingen, 72076 Tübingen, Germany; 2DWI—Leibniz Institute for Interactive Materials, Forckenbeckstr. 50, 52056 Aachen, Germany; 3Institute for Technical and Macromolecular Chemistry, RWTH Aachen University, Worringerweg 2, 52074 Aachen, Germany; 4University Eye Hospital Hamburg-Eppendorf, Martinistraße 52, 20251 Hamburg, Germany

**Keywords:** retina, drug-delivery, lipid-DNA nanoparticles, fluorophotometry, biodegradable, intravitreal injection, age-related macular degeneration, diabetic retinopathy, eye

## Abstract

Retinal eye diseases are the leading cause of blindness in the Western world. Up to date, the only efficient treatment for many retinal diseases consists of invasive intravitreal injections of highly concentrated drugs. Despite the fact that these injections are unpleasant for the patients, they potentially cause serious side effects, e.g., infections, bleeding within the eye or retinal detachment, especially when performed on a monthly basis, thus decreasing the injection frequency and lowering the desired drug dose. Therefore, a sustained released at the region of interest with a sustained release is desired. Recently, novel lipid-DNA nanoparticles (NPs) were shown to be an efficient drug delivery platform to the anterior segment of the eye. In this study, we investigated the distribution and tropism of the NPs when applied intravitreally, as a potential medication carrier to the posterior part of the eye. This technology is perfectly suited for the delivery of low molecular weight drugs to the back of the eye, which so far is greatly hindered by fast diffusion rates of the free drugs in the vitreous body and their intrinsically low retainability in ocular tissue. Excellent biodistribution, adherence and presence for up to five days was found for the different tested nanoparticles ex vivo and in vivo. In conclusion, our lipid-DNA based nanocarrier system was able to reach the retina within minutes and penetrate the retina providing potentially safe and long-term carrier systems for small molecules or nucleotide-based therapies.

## 1. Introduction

Retinal diseases are the leading cause of blindness in developed countries. The most common retinal diseases are age-related macular degeneration (AMD), diabetic retinopathy and retinal vein occlusion [1]. Due to the aging population, the number of patients suffering from these diseases is increasing. It is predicted that, by 2040, the number of AMD patients alone will have reached 288 million [2]. Treatment of these retinal diseases requires the injection of medication into the eye. Usually the application is intravitreal, but periocular application routes are sometimes used.

Drug delivery to the retina comes with a lot of obstacles and challenges. The first hurdle is the suitable application method. Local application is favored over systemic exposure, mainly due to the blood-retinal-barrier, which hinders substances to enter the eye from the bloodstream [3]. To reach the retina in the posterior segment of the eye, topical application would be desirable as it is non-invasive and easy to apply but crossing the anterior segment of the eye as well as distribution through the vitreous is severely hindered due to tissue characteristics and clearance mechanisms. Subretinal injections are the locally closest application, but the method comes with surgical risks and only targets a very small area of the retina directly at the injection side. Therefore, intravitreal injections is the most commonly used application method [4].

Once the substance is inside the eye tropism to the retina is the biggest obstacle as diffusion is hindered by the composition of the vitreous, especially the inner limiting membrane of the retina and the composition of the retina, as well as clearance mechanisms [5]. To increase the amount of drug at the target site drug-delivery vehicles are needed. These vehicles should facilitate tropism through the vitreous and retina, bypass clearance and preferably offer slow, but long-term drug release at the target site [6]. Various options have been explored to improve retinal delivery. In general, implants and degradable particles can be distinguished. Particles can be made from a variety of different materials and often several components are combined to achieve the best result. Apart from microspheres, liposomes, and nanoparticle formulations act as promising candidates for prospective drug delivery to the posterior segment of the eye [5]. Due to modifications in their molecular assembly properties, such delivery vehicles can be designed to match the respective demands or potentially even exceed them [7]. However, possible toxicity, accumulation of degradation products, difficulties in the formulation itself or in the loading capacities or simply the size may cause obstacles for clinical use [8].

DNA amphiphiles have been developed, consisting of oligonucleotides partially modified with alkyl chains, which self-assemble into micellar nanoparticles comprising a lipid core surrounded by a corona of single stranded DNA [9]. These lipid-DNA nanoparticles (NPs) have previously been applied for the anterior delivery of drugs to the eye [10]. They have been investigated as a drug-delivery vehicle for the topical treatment of glaucoma [11,12]. Furthermore, the successful application of the two antibiotics kanamycin and neomycin for the treatment of ocular infections using this delivery system has been shown [10].

This manuscript examines the potential use of these NPs as vehicles for the treatment of retinal diseases. For this purpose, two application methods are considered in more detail: Intravitreal and subconjunctival injections.

## 2. Materials and Methods

### 2.1. Materials

All chemicals and reagents were purchased from commercial suppliers and were used without further purification. Mydriatic agent was provided by the University Pharmacy Tübingen, with 9 mL containing: Phenylephrin-HCL (225 mg), Tropicamid (45 mg), Povidon 25 (270 mg) and Aqua ad. inj. (8.36 g).

### 2.2. Methods

#### 2.2.1. Preparation of Buffers and NPs

For the preparation of the Tris-buffered saline (TBS) buffer, 87.66 g NaCl (Fresenius Kabi Deutschland GmbH, Bad Homburg, Germany) and 60.57 g tris base (Roche, Penzberg, Germany) were dissolved in 800 mL of ultra-pure water and the pH was adjusted to 7.4 with 25% HCl. Ultra-pure water was added until reaching a total amount of 1 L of the 10× solution. For further use, a dilution of 1:10 in ultra-pure water was performed.

TRIS-acetate-Ethylenediaminetetraacetic acid (EDTA) (TAE) stock solution (2.5 M Tris-Acetate and 50 mM EDTA) was prepared by first dissolving 14.6 g EDTA (Sigma-Aldrich, Taufkirchen, Germany) and 242 g tris base in 800 mL of ultra-pure water while stirring. Afterwards, acetic acid was added to adjust the pH of the solution to 8.0. Finally, ultra-pure water was added to obtain a total volume of 1 L.

NP stock buffer was prepared by adding 100 µL of 50× TAE stock, 100 µL of MgCl_2_ (1 M) (Sigma-Aldrich, Taufkirchen, Germany) and 200 µL of NaCl (5 M) into an Eppendorf tube. Final concentration of NP buffer.

NPs were prepared in Eppendorf tubes in the desired amount of ultrapure water, NP buffer stock, the lipid oligonucleotide of interest and its fluorescently labelled complementary sequence bearing an Atto-488 dye on the 5′ end (biomers.net GmbH, Ulm, Germany). For the NP stock buffer 1.6 µL was added to 98.4 µL final solution to obtain a final concentration of 10 mM Tris-Acetate, 0.2 mM EDTA, 40 mM NaCl and 4 mM MgCl_2_, pH 8.0. The NPs were prepared at a concentration of 20 µM. Finally, NPs were formed and hybridized using a thermal gradient. In a thermocycler, the NP solutions were heated up and kept at (90 °C for the duration of 30 min). Afterwards, they were cooled down by 1 °C every two minutes until room temperature was reached. NPs were stored in the dark at room temperature until further use. 

Different DNA monomers were used to evaluate the influence of different DNA lengths and number of lipid residues [10]. In Table 1 an overview of the characteristics of the different NPs is given. The name of the respective NP already partially reveals the characteristics of the oligonucleotide. First, the letter “U” indicates the use of the lipid modified nucleotide. Next, the number following the U gives the total amount of modified deoxyuridine nucleotides in the sequence. The letter “T” in the name reveals that the modified bases are located on the terminus of the oligonucleotide, starting from 5′-end. Finally, the number in the end gives the total amount of nucleotides in the sequence.

#### 2.2.2. Fluorophotometric Measurements

In order to measure the fluorescence in the vitreous of the utilized eyes, an ocular fluorophotometer was employed (Fluorotron Master™, Ocumetrics, Mountain View, CA, USA) as described before [10,12]. Before measurements were performed, the eye of interest was aligned with the lens of the system and the optical axis of the Fluorotron Master™. The fluorescence was recorded in 0.25 mm sections along the determined optical axis by a program provided from Ocumetrics, Inc., which converts the reading to the corresponding concentration in ng/mL. The Fluorotron measures fluorescein concentration by counting photons at each step position, and computes the concentration using this formula: Fluorescein concentration (ng/mL) = photon count × Calibration Number × 1000/Gate Time (msec)/Ref Number − (Background Factor × 0.00001 × Calibration Number).

#### 2.2.3. Evaluation of Results Obtained from Fluorophotometric Measurements

In order to detect the NPs, they were functionalized with the fluorescent dye Atto-488, which was attached via covalent binding to the complementarily strand [10]. This strand was then hybridized to the NPs, thereby obtaining the fluorescently labelled vehicle. Binding of this dye to the DNA requires chemical modification of the structure of this molecule, which leads to alteration of its fluorescence strength. The magnitude of those changes is dependent on the sequence of the DNA it is coupled to. As the NPs constitute unique sequences (different length and different amounts of lipid modified bases), the dye exhibits specific intensities for each NP. This means that the fluorescence intensity measured for one NP cannot be directly compared to the next, or to the free dye. In order to make the found results for the NPs comparable, for each an individual starting value was defined which represents 100%. All the following measurements of the specific NP structure were divided by this first value, generating a relative value which can be compared between the different NPs.

For normalization two different starting values were utilized. The first normalization was performed by measuring the respective NP in solution within a cuvette before injection using the cuvette holder of the fluorophotometer. This value gives information about the total injected amount and which fraction is detected. It will from hereon be given as the “solution” measurement. The second one was measured directly after injection, which is displayed in the later figures as the “instant” time point. This value gives insight in the time dependent distribution of the NPs in the vitreous body. 

Here, analysis of variance (ANOVA) and Dunnett’s post-hoc tests were performed utilizing JMP 13. The small molecule control (Atto-488) data served as “control”. Significances are displayed in the graphs of the discussion as: * for the value of *p* ≤ 0.05, ** for the value of *p* ≤ 0.01 and *** for the value of *p* ≤ 0.001 [13].

#### 2.2.4. Intravitreal Injection of NPs into Ex Vivo Pig Eyes

Porcine eyes, freshly obtained from the local abattoir, were kept at a temperature of 4 °C and treated within 4 h after delivery. Before administration of NPs or control, the eyes were shortly washed with PBS and left to adjust to room temperature. For NP administration, the eyes were placed on the bottom of a petri dish and fixated with one hand. Afterwards, the pars plana was penetrated approximately 4 mm posterior to the limbus of the eye using a needle of 30 gauge (Hamilton Germany GmbH, Gräfelfing, Germany). Then, 40 µL of NP solution (20 µM) was injected into the center of the vitreous body under microscopic observation (Ophthalmology surgical retroskop opmi cs-xy S4, Zeiss, Oberkochen, Germany). The porcine eyes were then incubated for the designated time points in PBS.

For fluorophotometric evaluation, measurements were performed before injection and after 0, 5, 15, and 30 min as well as after 4, 8 and 24 h of incubation (n = 6–8).

For microscopy, the porcine eyes (n = 3) were embedded in Tissue-Tek and subsequently frozen in liquid nitrogen after 5, 15, 30 min, 1 or 2 h of incubation. Afterwards, cryosections were prepared, stained and imaged, as described in Section 2.2.7 [13].

#### 2.2.5. Intravitreal Injection of NPs into In Vivo Rat Eyes

Adult Lister Hooded rats were obtained from Charles River (Germany) and treated according to the German animal protection law (Research permission AK 3/11 and 1/15 issued by the Regierungspräsidium, Tübingen, Germany). Before treatment, the animals were anesthetized by peritoneal injection of a three components anaesthesia (0.1 mL/10 g bodyweight). For this, 5 mL of the anaesthesia was prepared using 0.5 mL Fentanyl (0.005 mg/kg) (Albrecht GmbH, Aulendorf, Germany), 2.0 mL Midazolam (2.0 mg/kg) (hameln pharma GmbH, Hameln, Germany), 0.75 mL Medetomidin (0.15 mg/kg) (Albrecht GmbH, Aulendorf, Germany) and 1.75 mL of 0.9% NaCl. Afterwards, the eyes were dropped with Conjuncain (4.0 mg/mL Oxybuprocainhydrochlorid, Bausch and Lomb GmbH, Berlin, Germany) further ensuring local anaesthesia, and a mydriatic agent (University Pharmacy Tübingen, −9 mL containing: Phenylephrin-HCL (225 mg), Tropicamid (45 mg), Povidon 25 (270 mg)) in order to increase visualization during microscopic observation of the position of the injection needle. For intravitreal application, 10 µL of NP solution (20 µM) was injected into a vitreous body, approximately 3 mm posterior to the limbus using a 30-gauge needle. 

After the surgery, an antidote was employed subcutaneously to terminate the anesthesia (0.3 mL/10 g bodyweight). For the preparation of 15 mL of the antidote, 1.5 mL naloxon (0.12 mg/kg) (hameln pharma GmbH, Hameln, Germany), 10 mL flumazenil (0.2 mg/kg) (Fresenius Kabi Deutschland GmbH, Bad Homburg, Germany) and 0.75 mL atipamezol (0.75 mg/kg) (Albrecht GmbH, Aulendorf, Germany) were added to 2.75 mL of 0.9% NaCl. Finally, gentamycin ointment (Ursapharm Arzneimittel GmbH, Saarbrücken, Germany) was applied onto the injection site to prevent infection.

For fluorophotometric evaluation, measurements were performed pre injection and 0, 1, 3 and 5 days post injection. In order to do so, the living rats were anaesthetized as described above and the eyes (n = 6) were dilated using the mydriatic agent. The anaesthesia was terminated by injection of the antidote.

For microscopy, the rats were euthanized with carbon dioxide inhalation after the designated time periods of 0, 0.5, 1, 3 and 5 days. The treated eyes (n = 6) were enucleated, frozen in Tissue-Tec (Sakura Finetek Europe B.V., Umkirch, Germany) and further processed as described in Section 2.2.7 [13].

#### 2.2.6. Injection of NPs into Periocular Tissue of In Vivo Rat Eyes

For these experiments, Adult Lister Hooded rats were treated and anesthetized as described in Section 2.2.5. Administration of 40 µL NP solution (20 µM) into the periocular tissue was performed subconjunctivally. A 30-gauge needle was inserted 5 mm into the conjunctival tissue approximately 10 mm temporolateral from the most lateral point of the limbus under microscopic observation. Antidote prepared as described above and administered at 0.3 mL/10 g bodyweight was administered subcutaneously to end the anaesthesia after surgery. In addition gentamycin eye drops were applied locally. The rats were euthanized with carbon dioxide inhalation after the designated time points of 30 min, 2, 4 and 24 h as well as three days. The treated eyes, including the surrounding subconjunctival tissue, were then excised, frozen in Tissue-Tec and further processed as described in Section 2.2.7. Throughout the different time points in this experiment, the analyzed eyes always equalled n = 2, except for the time periods of one and three days where, for U4T-12 and U6T-18 n = 6, eyes were examined [13].

#### 2.2.7. Fluorescent Microscopy and Imaging

Frozen sections of Tissue-Tek O.C.T. embedded porcine and rat eyes were longitudinally cut (12 µm) on a cryostat (Leica CM 1900, Leica Microsystems, Wetzlar, Germany), thaw-mounted onto glass slides (Superfrost plus, R. Langenbrinck Labor- & Medizintechnik, Emmendingen, Germany) and stored at −30 °C until further use. For visualization each section was fixed with 300 µL methanol (ten minutes) and afterwards washed with 1 × TBS solution (3 × 3 min). In a next step, the nuclei were counterstained with 100 µL 0.4 µg/mL 4′,6-Diamidino-2-phenylindol (DAPI) for five minutes, before being washed with 1 × TBS buffer (3 × 3 min) once more. After drying, the stained slides were covered in FluorSave (Calbiochem, Merck Millipore, Darmstadt, Germany) and furnished with cover glasses (R. Langenbrinck Labor- and Medizintechnik, Emmendingen, Germany). Finally, the prepared sections were imaged using a fluorescent microscope (Axioplan 2, Zeiss with Openlab software, Improvision, Coventry, UK). The nucleus stain was imaged employing a small band DAPI filter (wavelength of emission maximum: ~461 nm), whereas the fluorescent dye conjugated to the NP was captured utilizing a fluorescein-5-isothiocyanate (FITC) small band filter (wavelength of emission maximum: ~495/519 nm) [13].

## 3. Results

To assess the suitability of DNA nanoparticles as drug-delivery vehicles for the treatment of retinal diseases, the distribution and retention time of the NPs were first investigated ex vivo and then in vivo. In the ex vivo porcine eye, intravitreal application was used. The two NPs with the best performance were then used for in vivo investigations. In vivo, two application methods were compared: Intravitreal and subconjunctival injection.

### 3.1. Intravitreal Injection of NPs in Ex Vivo Pig Eyes

#### 3.1.1. Diffusion Properties of Various NPs in an Ex Vivo Vitreous Body

To investigate the diffusion and retention in the eye the fluorescently labelled NPs were injected intravitreally into ex vivo pig eyes. The fluorescence intensity of the different NPs was monitored for up to 24 h using an ocular fluorophotometer. By sequentially concentrating on spots along the optical axis (from retina to cornea) in the ocular cavity, as explained in Section 2.2.2, this device is able to quantify the intensity as a function of depth. These signals are then combined to create a fluorescence profile that is proportionate to the concentration of the NPs being employed (Figure 1). The first measurement was taken immediately following the NP injection. The signal was then tested at 5, 15, 30 and 1 h. The vitreous body received the strongest signal immediately following NP injection (Instant). Here, the maximum fluorescence intensity was at around 270 units. The NPs diffusion out of the ocular cavity was indicated by the signal’s continuous decline over time (Figure 1).

To quantify and compare the amount of NP present, the fluorescence intensity in the vitreous body at a given time point was integrated and normalized as explained in Section 2.2.3 for all eyes (n = 6–8). Free Atto-488 was employed as a small molecular control for the NPs. This data was normalized to both the fluorescence intensity of the solution prior to injection (“solution” value) and to the initial measurement made immediately after injections (“instant” value) in order to be able to directly compare this data with various NPs. The native fluorescence of the pig eyes used was also incorporated into the graph in order to compare the measured fluorescence upon injection to the situation prior to treatment (prescan).

Fluorescent measurements were taken for a period of 24 h following injection. Initial normalization was done to the “solution” value for the values (Figure 2A). While the other NPs only displayed fluorescence intensities between 19.0 and 39.1%, U6T-18 clearly demonstrated the highest retention in the eye with an immediate value after injection of 79.9% relative to the fluorescent intensity of the pure solution. The studied U6T-18 level remained at about 45.7% after an hour of incubation, which was more than twice as much as the closest-detected fluorescence for the other NPs (21.0% for U4T-12, 21.1% for U4T-18, and 20.6% for U6T-20).

Analyzing the NPs normalized to their respective solution value provided information about the detected fraction of NPs out of the total amount that was injected. A second normalization technique that placed more emphasis on the diffusion, dispersion, and retention of the NPs after injection was also employed. To achieve this, the second method included the fluorescence of each NP and conducted normalization to the corresponding instant measurement of the NP (Figure 2B).

Compared to the fluorescence directly after injection, U4T-12 had the best retention properties over time (Figure 2B). Out of the investigated NPs, only U4T-12 exhibited increased fluorescence intensity after the instant value, and had its maximum 30 min after injection. Two hours after injection, the fluorescence intensity for U4T-12 was still higher than the instant value and then slowly decreased over time. All other NPs exhibited a decreasing trend directly after injection with higher values than the control. Only U4T-18 displayed values below the ones of the control, five (78.0%) and 15 min (63.8%) after injection. After 8 h the flourescent intensity is 41.6% of the instant value for U4T-12 and 22.6% of the instant value for the other NPs, whereas the control is below 10%.

#### 3.1.2. NP Retention in the Vitreous and Adherence to the Retinal Tissue over Time

To further investigate the two best performing NPs (U4T-12 and U6T-18) in terms of retention time in the eye and adherence to the retina histological analysis were conducted. With regard to applying these NPs for drug delivery to the posterior segment adhesion to posterior tissue is of great importance and was therefore investigated by fluorescence microscopy. First, the control dye (pristine Atto-488) was injected into the pig vitreous body in order to prove that any observed adhesion of the NPs to the intraocular tissue is due to their structure. To this end, porcine eyes were injected with 40 µL of 20 µM Atto-488 solution. After defined incubation times of 5, 15, 30 min, 1 and 2 h in PBS the eyes were processed as cryosections, stained with DAPI and investigated under a fluorescence microscope (Figure 3). The eyes injected with the control did not show any positive results regarding adhesion of the fluorescent dye towards the retinal tissue in any of the analyzed time points. Additionally, in no other tissues or the vitreous body fluorescence was found in any of the cryosections. Hence, it can be concluded that the Atto-488 itself did not exhibit any affinity to the retinal tissue. 

In the next step, to investigate the affinity and tropism of the NPs to ocular structures, the further experiments were performed with the two best performing NPs. The retention of U4T-12 in the vitreous by examining the fluorophotometer measurements was analyzed (Figure 4A). The fluorophotometer data was normalized to the signal determined directly after injection. The integrated fluorescence intensity for the U4T-12 NPs increased after the first measurement to a maximum of 115.6% after 5 min. Relatively consistent fluorescence intensity was observed until the two hours measurement (112%). A sharp decrease followed and after 24 h approximately 15.6% of the initial fluorescence remained. 

Next, U4T-12 was injected in ex vivo porcine eyes and the adhesion was analyzed by microscopic fluorescence imaging (Figure 5B). In the histological pictures, U4T-12 NPs can clearly be identified as their green fluorescence is highly visible in the obtained images. The NPs formed a considerable layer on the inner parts of the retina, mainly on the inside of the ganglion cell layer (GCL), on the border to the vitreous body. At five minutes post injection, NPs could already be observed in the cellular structures. The green fluorescence was mainly found surrounding the DAPI stained nuclei on those cells which were facing towards the vitreous body. Strikingly, NPs could be observed that were not directly bound to the cells, but rather located in the vitreous body itself. This has not been the case for the control dye. Likewise, the NPs were fully covering the retina with the most pronounced adhesion site in the region of the outer GCL nuclei. For later time points, a more distinct line of the fluorescent NPs was visible with a dense pattern of the NPs located at the GCL. Additionally, at the later time points, the green fluorescence was found in more basal parts of the retina, especially one and two hours post injection. For example, at the two-hour time point, high coverage of the retina was found with an average penetration depth of the NP layer with around 120 µm. Obviously, the NPs penetrated through the GCL and reached the inner nuclear layer (INL). In almost all investigated eyes, the NPs were found to adhere to the cellular layers of the retina. In only one eye there were no NPs detected 30 min after injection. However, in later time points, good adhesion to the retinal structures was shown.

The fluorophotometric measurements revealed a good retention of U6T-18 in the vitreous body (Figure 5A). The integrated fluorescence intensity decreased to 78.0% after 5 min. Afterwards, the fluorescence intensity decreased in a constant manner over time, however always remained at a notably higher level than the control. Thirty minutes after injection the fluorescence intensity was still at 57.6% of the instant value. Two hours after injection, the detected intensity was still detected at 40.7%. After eight hours it remained just below 22.6%. 

In the U6T-18 histological sections of porcine eyes, the fluorescently labelled NPs were found covering the retina throughout the incubation time points with predominant adherence on the inside of the GCL (Figure 5B). Already five minutes after injection the fluorescent NPs were covering the entire GCL. In the 15 min image the NPs were observed as multiple spots sized below 80 µm in diameter, nevertheless displaying a strong intensity. The significant difference to the previously investigated NPs was the presence of NP fluorescence in the vitreous body, even after longer incubation periods. After one hour of incubation the coating was more intensive with an increased depth and covering approximately 90% of the retina. Still, at the two hour and last time point, the retention of U6T-18 was still detectable throughout the whole retina. 

These findings are also in good agreement with the observed results found in the fluorophotometric measurements. Additionally, the summarizing table (Table 2) underlined this with 14/15 eyes found to be positive. This was the same number of positive results that had also been found for the U4T-12 NPs.

In summary, the fluorescence imaging of U4T-12 and U6T-18 showed excellent coverage of NPs in regard to the retinal tissue. Both were found to be the only NPs out of this ex vivo setup displaying 14/15 positive eyes. Indicating superior performance, U4T-12 and U6T-18 NPs have been chosen for further in vivo experiments.

### 3.2. Intravitreal Injections of NPs in In Vivo Rat Eyes

#### Diffusion Profile and Adhesion Properties of the NPs in an In Vivo Rat Eye

In order to analyze the diffusion behavior of the NPs, in vivo rat eyes injected with the NPs were scanned using a fluorophotometer. However, unlike previously performed, the graphs were not integrated, as the in vivo rat eye model made the implementation of a constant measurement axis over several days extremely challenging. Representative scans of the injected Atto-488 control solution, U4T-12 and U6T-18 NPs, highlight the different diffusion manners of the control solution versus NPs (Figure 6A).

The autofluorescence observed in the eye is illustrated by the “pre injection” graph, which was obtained directly before injection. Several characteristic local peaks could be found, which indicated the position of the retina and the cornea. A smaller peak found between these two is due to auto fluorescence in the area of the iris and lens. 

As expected, a strong peak was detected right after injection of the control solution. The position of this peak indicated the location of the maximum concentration of the dye inside the vitreous. All the following measurements did not differ discernibly from the pre injection measurement, indicating rapid clearance of the control dye out of the optical axis of the eye. This was in good agreement with the prior results observed in the ex vivo experiments.

Compared to the detected fluorescence of Atto-488, the measured intensity for U4T-12 was approximately one fifth at the “0 days” measurement. This lower value can be explained by the conjugation of the dye to the complementary DNA, which changes the fluorescence properties of the dye. The position of the peak was similar to the one detected for the control in the vitreous. All measurements for the later time points did not display any positive peaks exceeding the levels of fluorescence found in the eye before injection. This finding was most likely due to changes in positioning of the eye between measurements and hence related to problems in aligning the optical axis of the eye to the optical axis of the fluorophotometer.

Concerning U6T-18, the detected fluorescence right after injection showed a different profile both in height and shape of the peak compared to the control. The detected fluorescence intensity was less than half the amount of Atto-488, which again could be attributed to the conjugation of the dye to the complementary DNA. Secondly, the main peak exhibited shoulder peaks on both sides. Nevertheless, the location of the peak was the same as the “0 days” peak found for Atto-488. Remarkably, one day after injection the peak had changed position and intensity. After three days, a slightly higher fluorescence was found than before injection. Here, the observed peak had shifted slightly towards the cornea. This observation presumably was due to aligning problems of the rat eye with the measured optical axis. Finally, five days after injection no difference to the initial state could be detected.

Furthermore, the adherence and tropism of Atto-488, U4T-12 and U6T-18 to the ocular structures were investigated by fluorescence microscopy (Figure 6B,C). Rat eyes were injected with 10 µL of the Atto-488 control (20 µM) into the vitreous. The rats were sacrificed after ½, 1, 3 or 5 days. Fluorescence imaging was used in order to define the adhesion sites and retinal tropism and the presence of the control. As expected, the eyes injected with the control were found to be negative in all investigated time points regarding the adhesion of the fluorescent dye to the retina. Likewise, no fluorescence was found in the vitreous body of any eyes. In accordance with the prior ex vivo experiments, it can be concluded that the small molecular control dye did not exhibit any affinity to the retina.

After ½ day, five out of six rat eyes were found positive for U4T-12. After day one, still four out of six eyes were identified as positive. After three days, this decreased to only half of the analyzed samples showing the presence of fluorescence and only one out of six eyes was positive for U4T-12 five days post injection (Figure 6C). 

The U6T-18 NPs revealed a lot of similarities in the ½ day time point compared to U4T-12 NPs. One remarkable difference was the NP stratum on the GCL, which was nowhere near as strong as for U4T-12. A couple of green spots indicated the presence of NPs between the GCL and the INL. A time-dependent manner in occurrence of positive findings was obvious for U6T-18 NP (Figure 6C). They were detected in five out of six eyes after ½ day and four out of six positive after one and three days. Only one out of six eyes was positive five days after injection. 

### 3.3. Injection of NPs into Periocular Tissue in In Vivo Rat Eyes

After demonstrating the retention in the vitreous and the adhesion potential of NPs after intravitreal injection in ex vivo and in vivo models, further experiments were conducted to investigate other possible sites of administration for the treatment of retinal diseases. One promising approach was to inject the drugs into the periocular tissue. From here, the drug could then diffuse into the ocular space and reach clinically relevant concentrations in the retinal tissue. 

Based on the positive results after intravitreal injection, the fate of the NPs in the periocular tissue was investigated in vivo using the fluorescence microscopy of cryosections. 

Since U4T-12 shows the strongest adhesion to ocular structures after intravitreal administration, this NP was selected for initial studies with subconjunctival injections. In this experiment, time points of 30 min, two and four hours were chosen and compared to the distribution of the small molecule control Atto-488. While these time points were significantly shorter than those chosen for intravitreal injection, they were chosen to provide preliminary information on the retention of U4T-12 in the periocular space.

As early as 30 min after subconjunctival application of Atto-488, the control did not provide positive results with respect to adherence to connective tissue outside the sclera (Figure 7A). No further fluorescence was detected in the cryosections either. 

In stark contrast, the animals treated with U4T-12 showed a large amount of the NPs in high intensity. A similar distribution of fluorescent NPs was seen for all time points. The U4T-12 NPs were mainly located on the outside of the sclera and formed a thick layer covering the entire scleral tissue. Thirty minutes after injection, the layer had a uniform thickness. In comparison, the average thickness varied more at the two later time points. In the image after two hours, the same tissue was covered, but the fluorescence was less intense. Similarly, the NP layer on the outside of the sclera was only about half as thick as at the 30-min time point. A similar observation could be made for the four-hour time point. There was no explicit fluorescence in the sclera, although the scleral tissue itself appeared to be infiltrated by the NPs, particularly in the 30-min image. It should be noted here that the U4T-12 carrier did not cross the sclera and therefore could not be found in the inner tissue of the eye.

Even after longer incubation times of U4T-12 and U6T-18, NPs signals could still be detected in the tissue. One day after injection, the U4T-12 NPs were found in the periocular tissue as strong green fluorescence on the outer sclera and connective tissue. The NP layer on the ocular surface showed a continuous layer throughout the tissue with a maximum thickness of about 10 µm.

The correlating U6T-18 micrographs showed less adherence of these specific NPs. The NPs outside the sclera appeared faint and diffuse but were still detectable. No measurements could be given due to the blurred boundaries. In addition, dozens of smaller fluorescent spots were observed on the outside of the conjunctival tissue under the conjunctiva. After three days, the displayed fluorescence looked very similar for both NPs. No continuous layer of NPs was visible.

In summary, the visible fluorescence decreased in strength and spatial distribution with increasing duration. Nevertheless, after one day of incubation, 5/6 eyes were positive for U4T-12 and at both time points for U6T-18. For U4T-12, as many as 6/6 eyes with NPs were detected in the subconjunctival tissue three days after injection (Figure 7B). However, no signs of fluorescence within the bulb were found for either NPs at any of the indicated time points.

## 4. Discussion

In recent years, considerable progress has been made in elucidating the pathological mechanisms of various eye diseases and also in their potential treatment. However, due to the special physiological barriers and the anatomical structure of the human eye, it is very challenging to deliver therapeutics to the desired site of action and current treatments suffer from low efficacy and non-specificity. Therefore, efficient drug delivery systems are needed to facilitate the delivery to the retina and prolong the retention time of drugs in the eye [4,14]. To this end, an increasing amount of nanotechnology-based carriers are being developed and applied to the treatment of ocular diseases. Due to their small size, combined with a large surface area, they can overcome barriers and be taken up by cells more easily than larger molecules, which allows them to increase the bioavailability of drugs [15]. The challenges in development include capabilities such as good biocompatibility, adhesion enhancement, targeted release, and increased longevity [16].

### 4.1. Variations of Adhesion Sites and in Adhesion Duration of NPs in the Ex Vivo Setup

To evaluate the distribution and amount of the NPs in the eye over time fluorophotometry was applied. Fluorophotometry revealed, that out of the investigated NPs only U4T-12 exhibited increased fluorescence intensity after the instant value, and had its maximum at 30 min after injection. For administration of the NPs, injections were performed at the limbus of the eye and the NP solution was injected in the approximate center of the vitreous body. However, the fluorophotometer only has a limited optical axis in which the NPs are detectable. Therefore, directly after injection the detected signal would have been lower than it should have been. In the course of the experiment the NPs might have diffused back into the optical axis of the fluorophotometer and the measured fluorescence could have increased thereafter. The process described might have resulted in a fluorescence maximum being later observed than instantly after injection. A second possible reason for the found increase is that injections could have been performed slightly outside the optical axis of the fluorophotometer. For measurement of the complete amount of injected NP, they would have first had to diffuse into the investigated tissues. This might have led to an increased fluorescent signal in the first measurements. Lastly, the observed phenomenon could also have been a combination of both explanations given [13]. 

Due to their superior performance, the U4T-12 and U6T-18 NPs were chosen for further in vivo experiments. After investigating the differences in diffusion for the different NPs, their adherence to ocular tissue and distribution was analyzed and compared by fluorescence microscopy. In order to summarize the previous observations of the fluorescence imaging Table 2 displays the number of positive findings out of the total number of eyes that have been investigated for the specific time point.

The first thing to notice is that, for the control out of the 15 investigated eyes, none were found positive. This strongly supports the efficacy of the investigated NPs. While the results of the fluorophotometry clearly indicated the presence of fluorescent dye directly after injection, no signs of fluorescence can be found in the pictures obtained by fluorescence microscopy. The explanation for this negative outcome could lie in the preparation of the cryosections. Before the evaluation under the microscope, each slide with the sections of the respective eye was fixated in methanol and washed six times with buffer. Due to the low adhesion capacity of the dye, it was expected that this leads to total clearance of the control from the ocular tissue. However, this also indicated the strong adhesion of the different NPs that were still to be found even after the same procedure was followed.

From the fluorescence images, it was clear that U4T-12 NPs showed the strongest adhesion to the retina compared to the other investigated NPs. Fluorescence could be observed reaching through the GCL into the INL in the one- and two-hour images, indicating strong adherence towards the retinal tissue. This was underlined by the 14/15 total positive eyes that have been found. Hence, it can be concluded that U4T-12 NPs presented exceptional properties regarding their adherence and distribution towards the retinal tissue. Christensen et al. showed that positively charged and PEG-coated liposomes could be retained at the retina for at least 24 h in ex vivo porcine eyes [17].

Recapitulating the findings for U6T-18, one must note the high predominance of fluorescent NPs which was still to be found in the vitreous body even for the longer periods of incubation. This clearly underlined the excellent retention qualities of this NP. Similar as for U4T-12, also for this NP 14/15 eyes were detected to be positive. However, the images showed that U6T-18 NPs also formed a constant coating displaying intense fluorescence on the GCL. This NP therefore seemed more favorable regarding adherence properties than U4T-18, U6T-12 and U6T-20 (Figure 2). Remarkably, these two NPs have the same percentage of lipid modified DNA compared to their total base-length (33%). This again indicates that the amount of modified nucleotides might play an important role for the adherence behavior of these NPs [13].

### 4.2. Examination of Adhesion Sites and Variations in Adhesion Duration of NPs in an In-Vivo Setup

For the in vivo examination of the adherence and distribution towards the retinal tissue, a summary of the positive findings is displayed in Table 3.

As already expected, the eyes injected with the control dye did not provide any positive results regarding adhesion of the fluorescent dye towards the retinal tissue. In contrast, U4T-12 and U6T-18 displayed consistent uptake of green in the retinal layers and the vitreous, especially for the shorter points in time. However, for both NPs, the same tendency of strong decrease in positive findings with increased incubation time could be observed. 

Interestingly, as seen before in the ex vivo images also for the in vivo experiments filamentous formations of NPs could be observed in the vitreous body. They were found for U4T-12 in the first two points in time and for U6T-18 at half a day and even five days. These formations were probably caused by adherence of the NPs towards the reticular framework of the vitreous composed of a network of type II collagen fibers. Interlaced into this 3D framework are glycosaminoglycans, hyaluronic acid and a variety of selected proteins. Further research is needed for identification of the structure responsible for the observed NP retention.

In conclusion, both examined NPs showed adhesion towards the retinal tissue for a maximum of five days. In more than 50% of the cases, NPs were found to be present after three days of incubation in fluorescence imaging. Additionally, the fluorophotometric analyses of U6T-18 confirmed the presence of NPs in the ocular tissue after three days of incubation. Other drug delivery systems have shown similar or even longer retention in the vitreous or the retinal tissue. This is a strong improvement compared to the retention of most drugs with low molecular weights, which have previously been shown to have an average half-life time of only several hours [18]. Kicková et al. were able to show, employing the pharmacokinetic simulations software STELLA^®^, that their pullulan-dexamethasone conjugates may release the loaded drug in the vitreous body over approximately 16 days in rabbit eyes and 25 days in human eyes. Employed in an in vivo setting the formulations were retained between three–five days in the rat vitreous [19]. Marano et al. showed that by applying a dendrimer drug delivery system intravitreally injected oligonucleotides against VEGF penetrated all of the retinal cell layers to the retinal pigment epithelium and were detectable for up to four months post injection in rats [20]. Sadeghi et al. examined the retention of liposomes manufactured out of a DSPC, DOPC, DSPG and DSPE-PEG mix in the vitreous body of rat eyes via fundus imaging for different formulations and could prove a retention period of up to 20 days in the case of the 1160 nm diameter formulations [21]. However, a drawback of liposomes is their tendency to lead to blurry vision due to their larger size [5]. A simple benefit of our NPs is that no blurry vision after implementation is likely, as they have an approximate diameter of 12 nm and their solution is clear. Nevertheless, also the NPs observed are subject to ocular clearance through the retinal tissues via passive and active transport mechanisms such as circulation of the aqueous humor, which may lead to excretion through the channel of Schlemm. However, it is expected that the NPs are detectable for longer periods of time than five days, but different analysis methods, like *Liquid chromatography*–mass spectrometry (LC-MS) for example, will be needed for this. The limiting factors in fluorescence imaging are restricted resolution and interference of the background fluorescence with fluorescence of NPs. Especially the latter one prevented more experiments with even longer times of incubation than for five days because with decreasing fluorescent intensity the NPs become harder to differentiate from the ocular tissue’s auto fluorescence [13]. 

## 5. Conclusions

The different evaluated NPs stayed concentrated in the vitreous body and retina of ex vivo pig eyes for 24 h, whereas the small molecule control quickly diffused out. Despite the slower diffusion of the NPs, they can already be found at the retina after 5 min of incubation, implying strong interactions with retinal tissue. In vivo experiments proved that the selected U4T-12 and U6T-18 NPs could be detected in the retina up to five days after injection, whereas the control was not visible directly after surgery. These results confirm the observations of the ex vivo setup. In conclusion, we confirmed the binding of our NPs to the retina in ex vivo and in vivo settings for several days. These promising results are inviting to conduct further research to test the efficacy of delivered medication and establishing the lipid-DNA NPs as a drug delivery platform for the posterior segment.

## 6. Patents

A.H., M.S.S., J.W.d.V. and S.S. are inventors of the presented technology. The patent (US10285939B2, EP3057572B1) is owned by the Medical Faculty of the University of Tübingen, Germany.

## Figures and Tables

**Figure 1 pharmaceutics-15-00532-f001:**
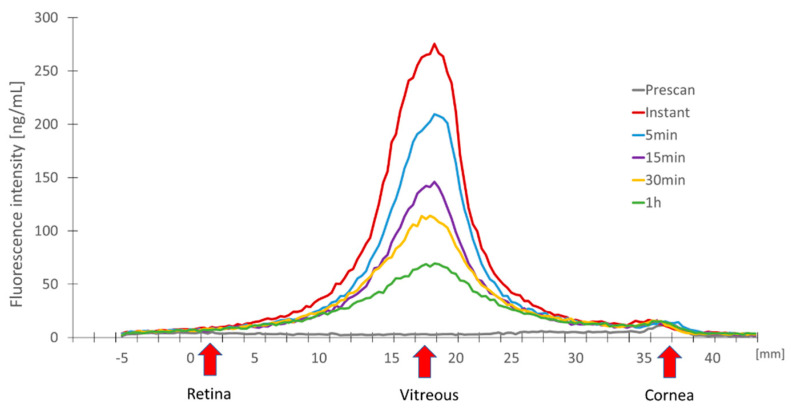
Fluorescence intensity (y axis) measured at different time points after intravitreal injection of U4T-18 NPs along the optical axis (x axis) in ex vivo porcine eyes. The decreasing intensity over time indicates diffusion of the NPs into the vitreous body. The red arrows indicate the localization of the retina, vitreous and the cornea.

**Figure 2 pharmaceutics-15-00532-f002:**
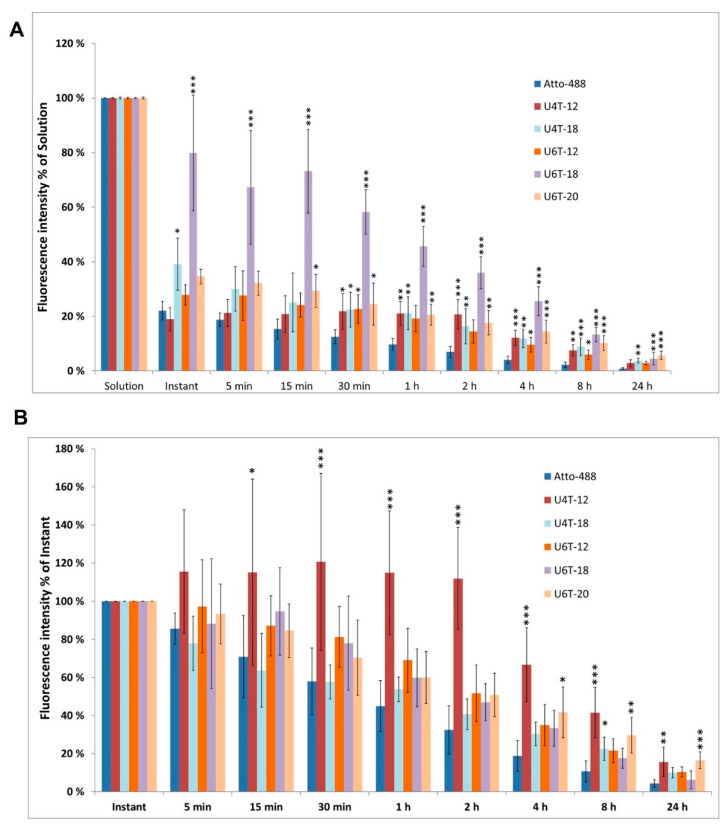
(**A**) Fluorescence intensity of different NPs normalized to the “solution” value. The strongest signal was observed with U6T-18. (**B**) Fluorescence intensity of different NPs normalized to the “instant” value. The strongest signal was observed with U4T-12. For both normalization methods the signal clearly decreased over time, indicating that the NPs are present in the vitreous body over prolonged periods of time. In contrast, the fluorescence intensity of the control was strongly decreased already after 15 min, indicating fast diffusion of the dye out of the vitreous body (n = 6–8). * indicate *p*-value of statistical analysis compared to the respective Atto-488 at the same time point, which are the values from “solution” for (**A**) and for “instant” for (**B**) * = *p* ≤ 0.05, ** = *p* ≤ 0.01, *** *p* = 0.001.

**Figure 3 pharmaceutics-15-00532-f003:**
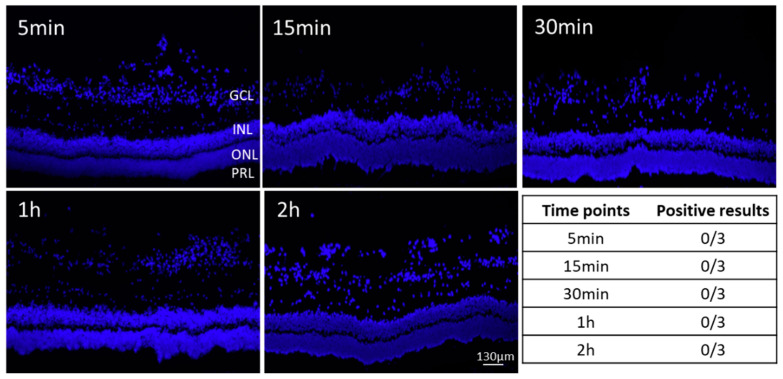
Representative fluorescent images of retinas of ex vivo porcine eyes after injection with the Atto-488 control and incubation for designated time points. No signs of fluorescence were observed. Nuclei are stained with DAPI (blue). Bottom right: Table with the number of control positive eyes at different time points, confirming the absence of the control dye. Abbreviations: Ganglion cell layer (GCL), inner nuclear layer (INL), outer nuclear layer (ONL), and photoreceptor layer (PRL).

**Figure 4 pharmaceutics-15-00532-f004:**
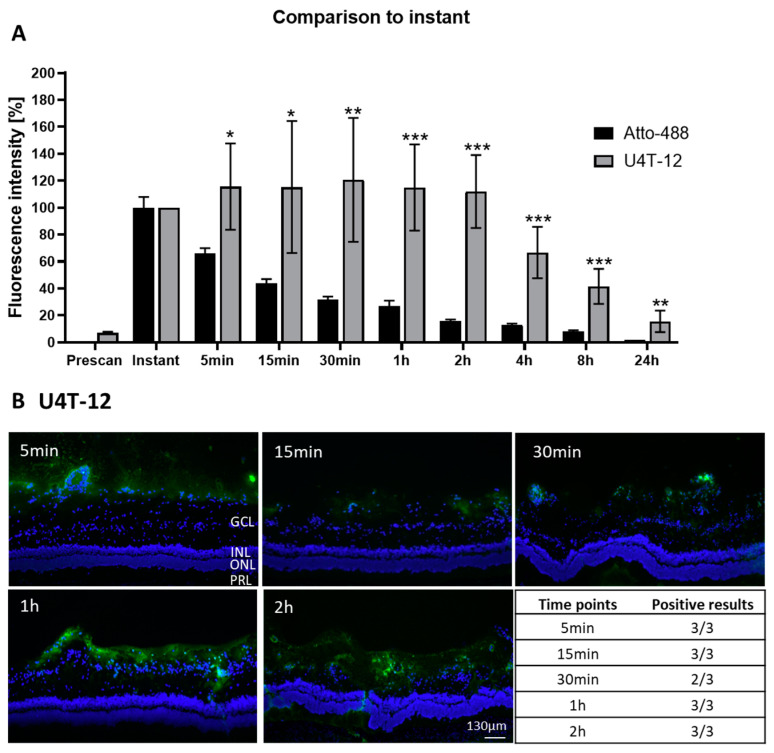
(**A**) Integrated fluorescence intensity normalized to the instant measurement at different time points after injection of the U4T-12 NPs (grey) into the vitreous body in ex vivo porcine eyes (n = 6–8), mean and standard deviation are shown. For comparison the data for the control Atto-488 is also given (black). U4T-12 displayed higher mean values at all observed time points compared to the control dye indicating excellent retention. * indicate *p*-value of statistical analysis (*t*-test) compared to the respective Atto-488 at the same time point); * = *p* ≤ 0.05, ** = *p* ≤ 0.01, *** = *p* ≤ 0.001. (**B**) Representative fluorescent images of retina (blue) of ex vivo porcine eyes, injected with U4T-12 NPs (green) and incubated for designated time points. Bottom right: Table with number of control positive eyes at different time points. The amounts of displayed NPs in the pictures and the results from the table indicated excellent adhesion towards the retinal tissues. Abbreviations: Vitreous body (VB), ganglion cell layer (GCL), inner nuclear layer (INL), outer nuclear layer (ONL), and photoreceptor layer (PRL).

**Figure 5 pharmaceutics-15-00532-f005:**
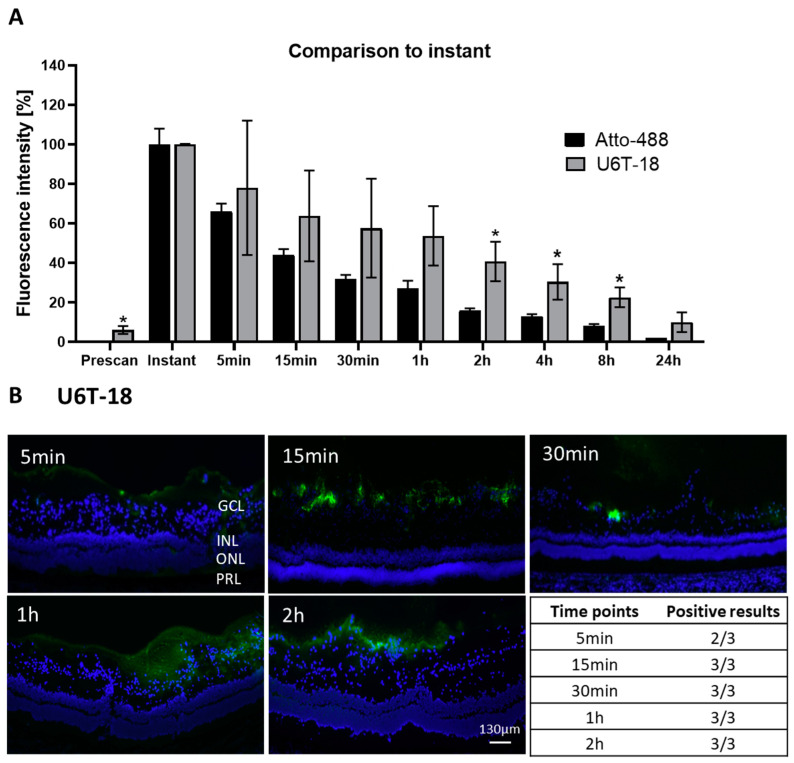
(**A**) Integrated fluorescence intensity normalized to the instant measurement at different time points after injection of the U6T-18 NPs (grey) into the vitreous body in ex vivo porcine eyes (n = 6), mean and standard deviation are given. For comparison, the data for the control Atto-488 is presented in the same graph (black). U6T-18 NPs exhibited higher fluorescence at all observed time points compared to the control dye, showing good retention. * indicate *p*-value of statistical analysis (*t*-test) compared to the respective Atto-488 at the same time point); * = *p* ≤ 0.05. (**B**) Representative fluorescent images of retina (blue) of ex vivo porcine eyes, injected with U6T-18 NPs (green) and incubated for designated time points. Bottom right: Table with number of control positive eyes at different time points. The amounts of displayed NPs in the images and the results from the table indicated excellent adhesion towards the tissues of the retina. Abbreviations: Ganglion cell layer (GCL), inner nuclear layer (INL), outer nuclear layer (ONL), and photoreceptor layer (PRL).

**Figure 6 pharmaceutics-15-00532-f006:**
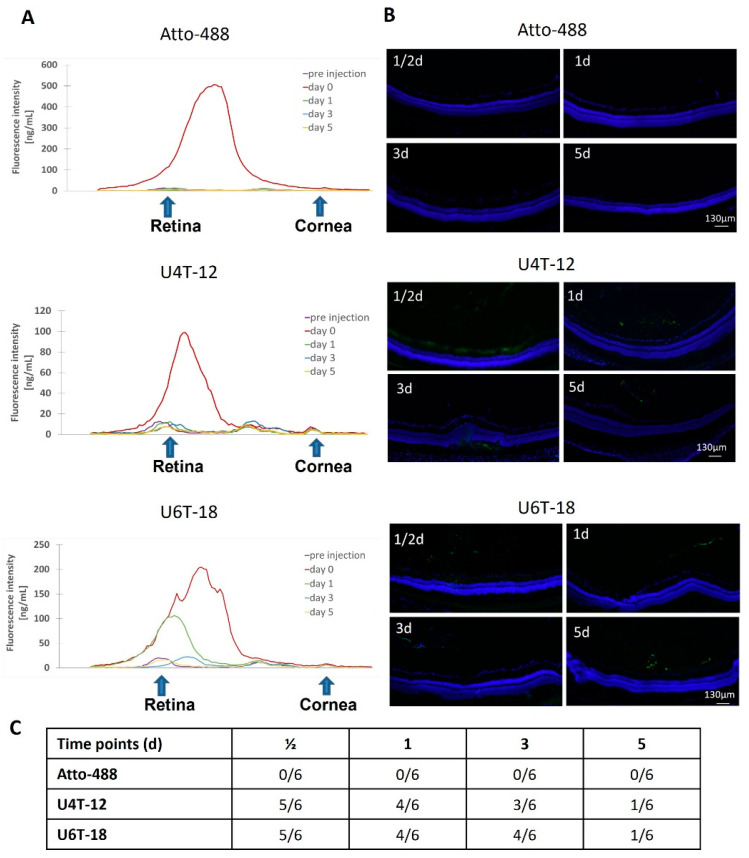
(**A**) Fluorescence intensity (y axis) measured at different time points after intravitreal injection of Atto-488, U4T-12 or U6T-18 along the optical axis (x axis) in an in vivo rat eye. Atto-488: The fast decrease in intensity over time indicated rapid diffusion of the dye into the vitreous body. U4T-12: The stark decrease in intensity on the first day can presumably be explained by difficulties in aligning the measured eye with the fluorophotometer. U6T-18: A drop in the intensity over time was observed, which stressed the retention properties of the NPs. (**B**) Representative fluorescent images of the retina (DAPI-staining: blue) of rat eyes injected with Atto-488, U4T-12 or U6T-18 (green) at different time points. (**C**) Table of positive results found in the eyes at different time points. In agreement with earlier experiments, no adhesion of Atto-488 towards the retina was found. Excellent adhesion of U4T-12 and U6T-18 towards the retina and retention in this matrix for up to five days.

**Figure 7 pharmaceutics-15-00532-f007:**
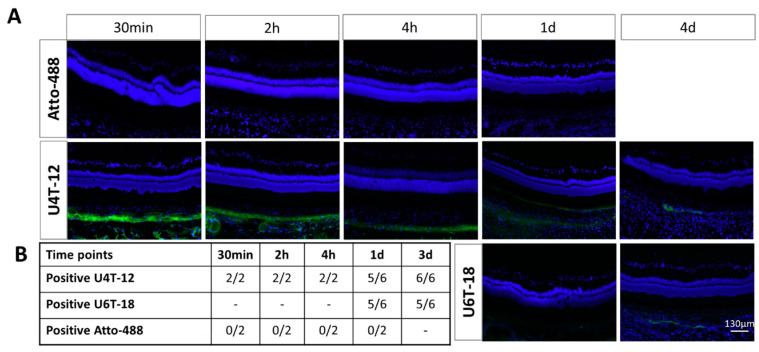
(**A**) Representative fluorescent images of rat eyes after periocular in vivo injection of U4T-12 NPs (short-term and long-term), U6T-18 (long-term) or pristine Atto-488. Visible are the retina, sclera and surrounding periocular connective tissue at different time points post injection. Nuclei are stained with DAPI (blue), Pristine Atto-488 and NPs are visible by green fluorescence. In stark contrast to the control dye, the animals treated with U4T-12 for 30 min, 2 h and 4 h displayed a large amount of the NPs at a high intensity. In the long-term experiments both U4T-12 and U6T-18 displayed efficient NP retention at the injection sites up to 3 days post injection, whereas Atto-488 did not show any retention at 1 day post injection. (**B**) Table summary of the long-term (U4T-18 and U4T-12) and the short-term (U4T-12) adhesion to the ocular structures after periocular injection.

**Table 1 pharmaceutics-15-00532-t001:** Sequence and details of lipid-modified oligonucleotides used for NP preparation. U represents the lipid-modified nucleotide. # indicates the number of lipid modified bases in total.

Name	Sequence (5′->3′)	Lipid Modified Bases # (%)
U4T-12	UUUUGCGGATTC	4 (33)
U4T-18	UUUUGCGGATTCGTCTGC	4 (22)
U6T-12	UUUUUUGGATTC	6 (50)
U6T-18	UUUUUUGCGGATTCGTCT	6 (33)
U6T-20	UUUUUUGCGGATTCGTCTGC	6 (30)

**Table 2 pharmaceutics-15-00532-t002:** Number of positive eyes found out of total number of investigated eyes by fluorescence microscopy after intravitreal injection of NPs or control in ex-vivo porcine eyes.

Point in Time	Atto-488	U4T-12	U6T-18
5 min	0/3	3/3	2/3
15 min	0/3	3/3	3/3
30 min	0/3	2/3	3/3
1 h	0/3	3/3	3/3
2 h	0/3	3/3	3/3
Total	0/15	14/15	14/15

**Table 3 pharmaceutics-15-00532-t003:** Number of positive eyes found out of total number of investigated eyes by fluorescence microscopy after intravitreal injection of U4T-12, U6T-18 or control in in-vivo rats. The numbers of positive findings out of all eyes examined are displayed at the bottom of the table. Additionally, here the U4T-12 and U6T-18 NPs displayed similarly superb results concerning the long-term adhesion and retention whereas the control did not exhibit any affinity to the retinal tissue.

Point in Time	Atto-488	U4T-12	U6T-18
½ d	0/6	5/6	5/6
1 d	0/6	4/6	4/6
3 d	0/6	3/6	4/6
5 d	0/6	1/6	1/6
**Total**	**0/24**	**13/24**	**14/24**

## Data Availability

Support data can be obtained via the corresponding author.

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
