# Peer review of "Lipid-DNA Nanoparticles as Drug-Delivery Vehicles for the Treatment of Retinal Diseases"

_pharmaceutics, 2023, doi:10.3390/pharmaceutics15020532_

Round 1

Reviewer 1 Report

Schnichels and coauthors investigated the distribution and tropism of intravitreally applied lipid-DNA nanoparticles as a potential drug carrier. Their results revealed that the lipid-DNA nanoparticles were able to reach the retina within minutes and penetrate the retina. I have comments and questions that may help to improve this manuscript as follows:

1) A few different terms were used in the manuscript for the test articles (lipid-DNA nanoparticles, DNA NPs, lipid-DNA based nanocarrier, lipid-modified DNA NPs, etc.). I recommend applying only one term corresponding to the nanoparticle structures used in this study.  

2) Please make sure all references were cited correctly in the introduction and discussion. Some references in the introduction and discussion are apparently cited only at the end of each paragraph, not in the specific sentence. The authors should give correct citations for each statement.  

3) The discussion is too long and the 4.3 (lines 624-707) is not directly related to their experiment. Since this is not a review paper, it should be revised focusing on comparison with the corresponding experimental results.

4) The first paragraph of the result (lines 228-232) is not necessary. It is a redundancy of the methods.

5) In line 242, “the signal was measured at 5, 15, 30 min and 2h.” However, Figure 1 showed 1h (not 2h). Please make it clear for the time point.

6) In figure 2, the indication of statistical significance is unclear. The authors noted that “*indicate p-value of statistical analysis compared to solution or instant”. They should be indicated with different symbols for comparisons between groups and at different time points.   

7) The authors noted that the steadily decreased fluorescence intensity in figure 1 indicates diffusion of the carrier from the ovular cavity. If so, how can be explained the increased fluorescence intensity of U4T-12 over time in figures 2, 4, and 5?

8) Quality of the figure 6A is poor. It should be replaced with a higher resolution (quality).  

9) Please ensure that all abbreviations are given full names the first time they are used.  

Reviewer 2 Report

Title:  Lipid-DNA nanoparticles as drug-delivery vehicles for the treatment of retinal diseases

Authors: Schnichels, S., Simmang, D., Löscher, M., Herrmann, A., Willem de Vries, J., Spitzer, M., and Hurst, J.

Summary: This paper describes the production and evaluation of lipid-DNA nanoparticles to deliver drugs to the retina for the treatment of a variety of retinal diseases such as age-related macular degeneration (AMD), diabetic retinopathy, and retinal vein occlusion. In this study they looked at several lipid-modified oligonucleotide nanoparticles to evaluate the efficiency of drug-delivery with different length oligonucleotides, as well as different DNA sequences. They evaluated two application methods, intravitreal and subconjunctival injections, for the treatment of retinal diseases. While nanoparticles have been investigated by others previously for the topical treatment of glaucoma to the anterior of the eye, in this study they evaluated the distribution and tropism of DNA nanoparticles when applied intravitreally as a drug delivery mechanism to reach the posterior part of the eye, which to date has been hampered by the fast diffusion rates of free drugs in the vitreous body, and low retainability in ocular tissue. In this study, there was excellent biodistribution, adherence and presence for up to 5 days for the nanoparticles tested providing a drug delivery mechanism to reach the retina and treat retinal diseases of the posterior compartment of the eye.

Revisions:

Page 1 – line 20: Change interested to interest.

Page 1 – line 29: Change save to salve.

Page 2 – line 45: Change barriers to barrier.

Page 2 – line 90: Need to include the molarity of the stock solution.

Page 3 – line 94: Need to include the molarity/final concentration.

Page 3 – line 127: What is the program used to convert the fluorophotometer reading to concentration?

Page 4 – line 158: Why were the eyes washed with ultra-pure water and not a saline buffer?   

Page 5 – line 195: The designated time periods based on the data shown should be 0, 0.5, 1, 3 and 5 days. The 0.5 time point is listed as 5.

Page 9 – Figure 4 legend, third line – change ans to and.  Remove the extra period after shown at the end of the sentence.  Under Abbreviations in the 2nd last line of the figure legend, after GCL, remove extra comma and then after PRL, remove the space before the period at the end of the sentence.

Page 10 – line 343: change hours to hour.

Page 11 – Figure 5 legend, fourth line, change (black). T U6T-18, change the T to The before U6T-18. Last line of this figure legend, the word outer is listed twice, remove one of these before the words nuclear layer.

Page 11 – Line 356: Change 30 to Thirty.

Page 11 – Line 363: Change minutes to minute.

Page 12 – Line 396: Change discernible to discernibly.

Page 13 – line 452: The word table should be capitalized, Table.

Page 14 – line 495: Correct the spacing of this line of text.

Page 15 – Figure 7, third line, capitalize table. 5th line, capitalized Long-term. 8th line, capitalize Table.

Page 15 – line 527: insert the word and after release, (before the word increased).

Page 16 – line 568: change hours to hour.

Page 18 – line 638: change choroidea to choroid.

Page 18 – line 641: remove parentheses around the word polyethylene glycol and polycaprolactone but keep the parenthesis around the abbreviations (PEG) and (PCL).

Page 18 – line 650: insert the word to after found and before be.

Page 18 – line 670: change proof to prove.

Page 19 – line 700: change proof to prove.

Page 20 – lines 744 – 753 – I don’t see any Appendix A or Appendix B, this looks like instructions to authors that was included in this paper and should be deleted.

Page 20 – 23, Reference section: The alignment for the references is not all flush and should be corrected to meet the formatting requirements of this journal.
